

# Development of a broadband cavity-enhanced absorption spectrometer for simultaneous measurements of ambient NO₃, NO₂, and H₂O

Woohui Nam[1], Changmin Cho[1], Begie Perdigones[1], Tae Siek Rhee[2], and Kyung-Eun Min[1]

[1]School of Earth Science and Environmental Engineering, Gwangju Institute of Science and Technology (GIST),123 Cheomdangwagi-ro, Buk-gu, Gwangju 61005, South Korea
[2]Korea Polar Research Institute, 26 Songdomirae-ro, Yeonsu-gu, Incheon 21990, South Korea

*Correspondence to*: Kyung-Eun Min (kemin@gist.ac.kr)

**Abstract.** We describe the characteristics and performances of our newly built broadband cavity-enhanced absorption
spectrometer for measurements of nitrate radical ($NO_3$), nitrogen dioxide ($NO_2$), and water vapor ($H_2O$). A customized
vibration-resistance cavity layout incorporated with $N_2$ purging on high-reflection mirror surfaces was implemented with a red
light-emitting diode (LED) as a light source. In general, this system achieved over 40 km (up to 101.5 km) of effective light
path length at 662 nm from a 0.52 m long cavity. For the accurate $NO_3$ measurement, the measured absorption spectrum of
$H_2O$ was used for simultaneous concentration retrievals with the other species, instead of being treated as interferences to be
removed or corrected prior to $NO_3$ detection. Synthesized $N_2O_5$ crystals under atmospheric pressure were used for performance
tests of linear response and transmission efficiency. From the standard injection experiments of $NO_3$, $NO_2$, and $H_2O$, high
linearities were observed ($R^2 \geq 0.9918$). The total $NO_3$ transmission efficiency through the system was determined to be 81.2 %
($\pm 2.9$, $1\sigma$) within the residence time of 2.59 seconds. The precisions ($1\sigma$) of $NO_3$, $NO_2$, and $H_2O$ in 1 Hz measurement from a
single pixel on the CCD were 1.41 pptv, 6.92 ppbv, and 35.0 ppmv with uncertainties of 10.8, 5.2, and $\geq 20.5$ %, respectively,
mainly from the errors in literature absorption cross-sections. The instrument was successfully deployed aboard the Korean
icebreaker R/V *Araon* for an expedition conducted in remote marine boundary layer in the Arctic Ocean during the summer
of 2021.

## 1 Introduction

The nitrate radical ($NO_3$) has drawn considerable attention due to its significant influence on nocturnal nitrogen oxide
chemistry after the first observation in the troposphere (Noxon et al., 1980; Platt et al., 1980; Brown and Stutz, 2012). $NO_3$ is
mainly produced from the oxidation of nitrogen dioxide ($NO_2$) by ozone ($O_3$, R1) and is in thermal equilibrium with dinitrogen
pentoxide ($N_2O_5$) from its further combination reaction with $NO_2$ (R2).

The role of $NO_3$ as an oxidant especially for the unsaturated volatile organic compounds (VOCs) becomes more critical at
night not only because of its extremely low abundance due to the losses by rapid photolysis (R3; Stark et al., 2007) and reaction



with NO (R4) but also because of the negligible amount of photochemically induced hydroxyl radical.  Particularly, alkenes
from biogenic sources with more than two double bonds (e.g., isoprene and terpenes; Winer et al., 1984; Ng et al., 2017) and
reduced sulfur compounds like dimethylsulfide (DMS; Allan et al., 2000) are susceptible to be oxidized by $NO_3$ (R5).

$NO_2 + O_3 \rightarrow NO_3 + O_2$ (R1)

$NO_3 + NO_2 + M \leftrightarrow N_2O_5 + M$ (R2)

$NO_3 + h\upsilon \rightarrow NO_2 + O(^3P)$ (< 587 nm)

$NO_3 + h\upsilon \rightarrow NO + O_2$ (< 714 nm) (R3)

$NO_3 + NO \rightarrow 2NO_2$ (R4)

$NO_3 +$ alkenes, aromatics, DMS $\rightarrow$ product (R5)

$N_2O_{5(g)} + (H_2O_{(aq)}$ and/or $Cl^-_{(aq)}) \rightarrow (2 - \varphi)NO^-_{3(aq)} + \varphi ClNO_{2(g)}, (0 \leq \varphi \leq 1)$ (R6)


As shown in reaction (R6), reservoir species of $NO_3$, $N_2O_5$, can undergo heterogeneous reaction to form gaseous nitryl
chloride ($ClNO_2$) and/or aqueous nitrate (Roberts et al., 2008). This uptake on aerosol can act as a dominant nitrate formation
path under haze events (Chang et al., 2018; McDuffie et al., 2019; Lin et al., 2020; Liu et al., 2020). Meanwhile, $ClNO_2$ can
be photolyzed to $NO_2$ and Cl radical after following sunrise; consequently, it does not only conserve $NO_x$ (=NO + $NO_2$) but
also accelerate oxidation speed by adding the Cl radical into the atmosphere (Osthoff et al., 2008; Le Breton et al., 2018).

In addition, the important roles of daytime $NO_3$ and $N_2O_5$ have been reported in terms of their contributions to VOC
oxidation and aerosol evolution (Geyer, 2003; Brown et al., 2005; Osthoff et al., 2006; Wang et al., 2014; Brown et al., 2016;
Brown et al., 2017; Wang et al., 2020; Foulds et al., 2021). Brown et al. (2017) observed non-negligible amount of $N_2O_5$ (up
to 35 pptv) during the daytime under high $NO_x$ conditions and Foulds et al. (2021) found competitive $NO_3$ loss by VOC
oxidation with photolysis, even in daytime. These findings indicate that the impacts of $NO_3$-driven chemistry is not limited to
nighttime.

Because of high reactivity, thus short lifetime, and low mixing ratio (a few to several pptv), observation of the ambient $NO_3$
is challenging. To our best knowledge, no commercial instruments are available at present, and only a few in situ measurement
techniques have been used. Systems based on laser-induced fluorescence (LIF; Wood et al., 2003; Matsumoto et al., 2005) and
matrix isolation electron spin resonance (MIESR; Mihelcic et al., 1993) had been reported. However, most of the contemporary
instruments in active use even in intercomparison work (Dorn et al., 2013) are based on absorption spectroscopic techniques
capturing the strong $B^2E'$-$X^2A'_2$ electronic transition of $NO_3$ at 662 nm (Yokelson et al., 1994). Differential optical absorption
spectroscopy (DOAS), characterized with a long physical path length, has been widely used for several decades (Platt et al.,
1980; Heintz et al., 1996; Allan et al., 1999; Geyer et al., 2001b; McLaren et al., 2004; Stutz, 2004; Vrekoussis et al., 2004; Li





et al., 2007; Sommariva et al., 2007; Asaf et al., 2009; Wang et al., 2013; Lu et al., 2016). Meanwhile, more compact instruments with an optical cavity have been recently developed for $NO_3$ measurements; cavity ring-down spectroscopy (CRDS; King et al., 2000; Brown et al., 2001; Simpson, 2003; Ayers et al., 2005; Nakayama et al., 2008; Schuster et al., 2009; Flemmer and Ham, 2012; Hu et al., 2014; Wang et al., 2015; Li et al., 2018; Wu et al., 2020) and cavity-enhanced absorption spectroscopy (CEAS) are the most representative techniques. In particular, CEAS instruments with broadband light sources

for $NO_3$ measurement were developed for field (Langridge et al., 2008; Varma et al., 2009; Kennedy et al., 2011; Wang et al., 2017; Suhail et al., 2019; Wang and Lu, 2019) and laboratory studies (Venables et al., 2006; Wu et al., 2014; Fouqueau et al., 2020) with advantages of capability for simultaneous measurements of multiple species and applicability of cheap light sources (e.g., light-emitting diode (LED), arc Xe lamp, and supercontinuum radiation source).

One of the main difficulties for accurate $NO_3$ measurement by CEAS is $H_2O$ treatment. Due to the strong but narrow

absorption by $H_2O$ around 660nm (Ball and Jones, 2003), the contribution of $H_2O$ on light extinction needs to be well characterized. Several methods such as look-up table (Langridge et al., 2008; Varma et al., 2009; Suhail et al., 2019), iterative calculation (Kennedy et al., 2011), and frequent $NO_3$ zeroing via NO titration (Wang et al., 2017) were adopted to overcome this issue.

Based on the capability of species-specific absorption cross-section measurements by CEAS (Axson et al., 2011; Chen and

Venables, 2011; Young et al., 2011; Kahan et al., 2012; Sheps, 2013; Thalman and Volkamer, 2013; Young et al., 2014; Prakash et al., 2018; He et al., 2021; Wang et al., 2022), we suggest a new approach using measured $H_2O$ spectrum for simultaneous quantification of $NO_3$, $NO_2$, and $H_2O$ which is simple and efficient enough for atmospheric application. Through this manuscript, we present not only our newly built broadband cavity-enhanced absorption spectrometer (BBCEAS) with detailed descriptions of design and performances but also the linearity test results with $H_2O$ in atmospherically relevant ranges.

Moreover, we also show the results of the shipborne measurement of $NO_3$ in the Arctic Ocean, indicating successful performance of the instrument in field application.

## 2 Instrumental setup

BBCEAS is a sensitive technique to directly measure the abundance of target species and/or its optical extinction properties, introduced by Fiedler et al. (2003). Details on the working principle can be found in Ball and Jones (2003) as well as Gagliardi

and Loock (2014). Briefly, this technique is based on the measurement of light extinctions in a relatively broad wavelength range. System based on this method basically consists of a broadband light source, a high-finesse optical cavity formed by a pair of high-reflection (HR) mirrors, and detector(s). The light from the source resonates inside the cavity and rapidly reaches a steady state with the wavelength-specific attenuated intensity through the loss processes of (1) transmission, diffraction, and absorption by HR mirrors, (2) scattering by particles and gases (i.e. Mie and Rayleigh scattering), and (3) absorption by

sampled trace gases in the cavity. The transmitted light from the cavity is then detected by the spectrometer and charge-coupled device (CCD) or photodiode array to monitor the light extinction spectrum.



The schematic diagram of our single channel BBCEAS is shown in Figure 1. The material, dimensions, and design of the novel cage system on optical mounts for stable performance with respect to vibration and pressure changes during the field deployment were adopted from the glyoxal (CHOCHO) and nitrous acid (HONO) instrument described in Min et al. (2016).

Detailed descriptions of the optical layout, flow system, and data acquisition system are described in the following section.

## 2.1 Optical layout

An LED (LZ1-10R202-0000, LedEngin, Germany) centered at 660 nm is used as a broadband light source and is mounted on a home-built module with copper plate, thermoelectric cooler, heatsink, XYZ translator (LP-1A-XYZ, Newport Corp., USA), and fan to control the position and temperature ($20\pm0.1$ °C) precisely. The light from the LED is collimated by an off-

axis parabolic mirror (50328AU, Newport Corp., USA) and enters the cavity. The HR mirrors (FiveNine Optics, USA) for high-finesse optical cavity are mounted at a distance of 51.5 cm occupied by a Teflon cell (1 in. outer diameter (o.d.), and 1 mm thickness). The light exiting the cavity is then focused by another parabolic mirror (50331AU, Newport Corp., USA), filtered through a colored band-pass filter ($660\pm5$ nm, FB-660-10, Thorlabs Inc., USA), and coupled to a fiber collimator (74-UV, Ocean Optics Inc., USA). The customized optic fiber (Seokwang Optical Co., Korea) with linearly assembled seven 200

μm diameter cores are aligned along the slit axis of the spectrometer (HRS-300MS-NI, Princeton Instruments Inc., USA).

Light is then transmitted to a diffraction grating (1200G/mm, 750nm blaze, 68 mm × 68 mm) and dispersed with respect to its wavelength. A CCD is used as a detector (PIXIS-2KX, Princeton Instruments Inc., USA) to monitor the spectra of the final transmitted light intensity, and is cooled to -70 °C to minimize the dark current. The wavelength coverage from 632 to 691 nm and the spectral resolution of 0.47 nm as full width at half maximum (FHWM) were calibrated with the narrow Ne emission

lines (NE-2, Ocean Optics Inc., USA).

The overall configuration of the optics is similar to the system described by Min et al. (2016); however, the main difference is that we have an independent cavity for individual channel rather than two channels sharing one parabolic mirror plate. This modification minimizes potential interferences of light leaking from the adjacent cell and provides convenience in operation and maintenance aspects (i.e. HR mirror cleaning and LED swapping).

## 115 2.2 Flow and data acquisition system

Air sample is drawn into perfluoroalkoxy alkenes (PFA) inlet tubing with a constant flow rate of 2.5 slpm (standard liter per minute) by a mass flow controller (Alicat Scientific, USA) and a scroll pump (IDP-3, Agilent Technologies Inc., USA). The design of the coaxial inlet following Min et al. (2016) is used to minimize the pressure change during the mirror reflectivity measurements and sampling cycles (see Sect. 3.1.1). On the downstream of the inlet, a 2μm polytetra-fluoroethylene (PTFE)

membrane filter (R2PJ047, Pall Corporation, USA) is used to minimize the light extinction owing to Mie scattering by sampled aerosols inside the cavity as well as to prevent reflectivity degradation of HR mirrors. The air then passes through 1/16 in. inner diameter (i.d.) PFA tubing into the cavity to minimize $NO_3$ loss by shortening the residence time with the scheme of





reduced pressure operation as in Fuchs et al. (2008). The temperature and pressure inside the cavity are measured at the outflow of the cell.

Another difference with the cavity system in Min et al. (2016) is the addition of purging system. As an active strategy to prevent sampled air contact on HR mirror surface, ultra-high purity (UHP) $N_2$ flows (> 99.999 %, 20 standard cubic centimeter per minute, sccm) are introduced on each side of the cavity mirrors via custom-designed PFA cell flanges with orifices (50μm, SS-1/8-Tube-50, Lenox Laser, USA). The effect of purge flow on the volume occupied by the air sample in the cavity was estimated as in Sect. 3.1.2.

Our instrument is operated and controlled automatically with the customized software programmed by LabView (National Instruments) for flow rates and temperatures of LED and CCD. The transmitted light spectra and other auxiliary data including temperatures, pressures, and flow rates for the extinction spectrum calculation are acquired via this program as well.

## 3 Characterization

    From the spectrum of measured light extinction inside the cavity, $\alpha(\lambda)$, the number density ($x_i$) of species $i$ can be
calculated from the equation (1).

$$\alpha(\lambda) = \sum T_i \sigma_i(\lambda) x_i + \alpha_{Mie} + \alpha_{Rayleigh} = \left(\frac{1-R(\lambda)}{d_{eff}}\right)\left(\frac{I_{out,0}(\lambda)}{I_{out}(\lambda)} - 1\right) \tag{1}$$

Here, $T_i$ is the transmission efficiency from the inlet to the cavity, $\sigma_i$ corresponds to the absorption cross-section, $R$ stands for mirror reflectivity, and $d_{eff}$ is effective cavity length. The $\alpha_{Rayleigh}$ and $\alpha_{Mie}$ refer to the optical extinctions due to Rayleigh and Mie scattering, while we neglect $\alpha_{Mie}$ owing to the aerosol filter on the inlet airway. The intensities of light transmitted
from the cavity with and without the absorbing species are symbolized as $I_{out}$ and $I_{out,0}$, respectively. Here, we define $I_{out,0}$ as the light intensity when the cavity is filled with dry zero air (ZA) only. Thus, for the accurate quantification of $x_i$ based on equation (1), not only the reference spectrum $\sigma_i$ but also the instrumental parameters such as $d_{eff}$, $R$, $I_{out,0}$, and $T_i$ should be characterized beforehand.

### 3.1 Determination of cavity parameters

145       **3.1.1 Mirror reflectivity, $R(\lambda)$**

  $R(\lambda)$ can be derived from the well-known Rayleigh scat

  tering differences in two species and we selected helium (He) and ZA as shown in equation (2). For our instrument, 2.75 slpm flow of He or ZA (99.999 % each, Daedeok Gas Co. Ltd.) is overflowed into coaxial inlet tubing so that 2.5 slpm flow is introduced into the cavity same as the flow rate of sample, while the rest of the flow is streamed towards the outside of the
inlet to minimize the pressure change inside the cavity.

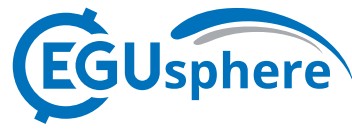

$$R(\lambda) = 1 - d\left[\frac{I_{ZA}(\lambda)\alpha_{Rayleigh,ZA}(\lambda) - I_{He}(\lambda)\alpha_{Rayleigh,He}(\lambda)}{I_{He}(\lambda) - I_{ZA}(\lambda)}\right] = 1 - \frac{d}{L_{Light}(\lambda)} \qquad (2)$$

In equation (2), $I_i$ refers to the transmitted spectral intensity filled with gas species $i$. $L_{Light}$ is the theoretically calculated effective light path length under the assumption that light attenuation is solely driven by the mirror and the cavity length, $d$. The $\alpha_{Rayleigh,i}$ was calculated from the literature Rayleigh scattering cross-sections of $N_2$ (Bodhaine et al., 1999), $O_2$, and He

(Shardanand and Rao, 1977) considering pressure and temperature changes.

Figure 2 shows the cavity characteristics such as the $I_{ZA}(\lambda)$, $I_{He}(\lambda)$, $R(\lambda)$, and $L_{Light}(\lambda)$ as examples acquired during the $T_{E,NO3}$ quantification experiments (described in Sect. 3.3.2). Although the difference of the transmitted light intensities between He and ZA injections was largest at near 662 nm (Figure 2 (a)), the wavelength corresponding to the highest $R(\lambda)$ value was shifted to 672.5 nm due to the incorporated wavelength-dependent characteristics of Rayleigh scattering and mirror

reflectivity (Figure 2 (b)). $L_{Light}$ exceeded 98 km (Figure 2 (c)) within the wavelength range for fitting (shaded in light red in Figure 2). Especially at 662 nm where $NO_3$ absorption peaks, $R$ and $L_{Light}$ were 99.9995 % and 101.5 km, which are superior to other reported ones from previous studies (Table 1 in Sect. 3.5). The uncertainty for $R(\lambda)$ estimation is 2.2 %, mainly from error in Rayleigh scattering spectrum of ZA (2 %; Washenfelder et al., 2008) and rarely from the temperature and pressure measurements (0.7 and 0.5 %, respectively).

**3.1.2 Effective cavity length, $d_{eff}$**

The volume occupied by the sample in the cavity should be accurately evaluated. Due to the reduction in the sample volume by addition of the purge flow, the $d_{eff}$ differs from both the cavity length defined by mirror separation (51.5 cm) and the physical displacement of sample in and out ports (47.5 cm). This parameter has been commonly quantified by the injections of known amounts of standard gas such as $H_2O$ (Kennedy et al., 2011), $O_3$ (Dubé et al., 2006; Fuchs et al., 2008), and $NO_2$

(Schuster et al., 2009; Wang et al., 2017). In this study, we filled the cavity with 5 ppmv $NO_2$ standard (in $N_2$, Korea Research Institute of Standards and Science, KRISS) and compared the retrieved $NO_2$ number densities with and without the purge flow. From this experiment, we concluded that the purge volume takes 1.67 cm$^3$ in the cylindrical cell (1 in. o.d. and 1 mm thickness) and thus estimated $d_{eff}$ as 50.28 cm with an uncertainty of 5.2 % (1 % from the standard gas, 2.9 % and 2.6 % from retrieval errors, and 3.2 % from $\sigma_{NO2}$ reported by Bogumil et al. (2003)).





**3.2 Retrieval of number density**

### 3.2.1 Absorption cross-section, $\sigma_i$

Number density $x_i$ is determined from the optimal fit of $\sigma_i(\lambda)$ on the light extinction spectrum, $\alpha(\lambda)$, within broad spectral ranges. For $\sigma_{NO_3}$, convolved spectrum of literature cross-section from Yokelson et al. (1994) was used because of its simple outstanding absorption features as shown in Figure 3. Its uncertainty was reported as 10 %.

Measured spectra were used for $H_2O$ and $NO_2$ to compensate for the imperfections in line-shape determination. These could be originated from the astigmatic bias on CCD pixels and from the environmental changes (i.e. pressure and temperature) in instrumental operation conditions. In order to minimize the impacts of these accumulated errors on concentration retrievals, measured spectra were used in previous studies (Min et al., 2016; Liang et al., 2019; Barbero et al., 2020) based on the fact that the CEAS technique is widely used to characterize the wavelength-dependent light extinction properties of chemical

species (Thalman and Volkamer, 2010; Axson et al., 2011; Chen and Venables, 2011; Young et al., 2011; Kahan et al., 2012; Sheps, 2013; Thalman and Volkamer, 2013; Young et al., 2014; Prakash et al., 2018; Jordan et al., 2019; He et al., 2021; Wang et al., 2022).

    As mentioned before, the treatment of $H_2O$ absorption features is crucial for accurate $NO_3$ measurement due to its strong but narrow absorption lines around 660 nm. To prevent this issue, the measured $H_2O$ spectrum scaled with literature and

relative humidity (RH) probe was used as $\sigma_{H_2O}$. $H_2O$ was produced by flowing ZA through a deionized water bubbler at room temperature, while activated carbon denuder (6 mesh, Ecotech Pty Ltd., Australia) as well as Drierite filter (8 mesh, Thermo Fisher Scientific, USA) were installed on upstream of the bubbler to remove possible contaminants in ZA cylinder. An averaged $H_2O$ spectrum for 15 minutes injection was scaled with the literature spectrum from HITRAN2020 database (Gordon et al., 2022) in the range of 659.28-671.94 nm and the humidity transmitter (HMT337, Vaisala, Finland, uncertainty : 1 %)

data.

    Due to the weak but complex absorption features of $NO_2$ in the fitting range, we measured the absorption spectrum produced from 3-minute injection of $NO_2$ standard gas (10 ppmv in $N_2$, uncertainty 1 %, KRISS) passing through a trap immersed in dry ice to ensure $H_2O$-free condition. The acquired spectrum was scaled to the literature one from Bogumil et al. (2003) for the range of 649.64–672.79 nm where the most apparent $NO_2$ absorption features exist with minimum fitting error.

Although the application of the measured spectra improves the fitting performance in general, the uncertainties of $\sigma_{H_2O}$ and $\sigma_{NO_2}$ inevitably increased ($\geq$20.1% and 3.3%, respectively) compared to the literature's ($\geq$20% and 3.2%) owing to the additional errors caused by fitting procedure (1.3 % and 0.036 %) and number density calculation (0.86 % for both). Note that we only provide the lower limit error in $\sigma_{H_2O}$ propagated from the absorption line-by-line uncertainties in HITRAN2020 ($\geq$20%).





### 3.2.2 Spectral fitting


The Levenberg-Marquardt least-squares fitting software, DOASIS (Kraus, 2006), was applied for the spectral fitting of the extinction spectrum between 659.28 and 671.94 nm. Fourth-order polynomial was applied to account for the optical drift and/or unaccounted extinctions such as absorption by ambient ozone. The reference spectra were allowed to be shifted within $\pm 1.0$ nm and squeezed freely for the $\sigma_{NO_3}$ but the measured ones were set to share the degree of horizontal shift and squeeze

together. Figure 4 shows an example of simultaneous retrievals of 5.45 pptv $NO_3$, 5.75 ppbv $NO_2$, and 5620 ppmv $H_2O$ with polynomial and fit residual from the ambient measurement during the Arctic shipborne mission (acquired at the open ocean on 26 August 2021, 17:11:41 UTC described in Sect. 4) with 2 seconds integration time.

### 3.3 NO₃ wall loss evaluation

Even though CEAS is able to measure the target species without concentration calibration using chemical standards in regular

operation, indeed, the loss of $NO_3$ along the airway needs to be evaluated. Many previous works carefully characterized transmission efficiency of $NO_3$ for their instruments (Aldener et al., 2006; Dubé et al., 2006; Fuchs et al., 2008; Schuster et al., 2009; Kennedy et al., 2011; Wagner et al., 2011; Hu et al., 2014; Wang et al., 2015; Sobanski et al., 2016; Wang et al., 2017; Li et al., 2018). In line with this, we conducted a series of $NO_3$ injection experiments with a custom-built $NO_3$ generation system.

### 3.3.1 NO₃ generation


$NO_3$ was delivered from the synthesized $N_2O_5$ crystal under atmospheric pressure with the home-built system (Figure 5). To produce gaseous $N_2O_5$ via reactions (R1) and (R2), 5 to 10 % of $O_3$ was generated by corona discharge (Nano 15, Absolute Systems Inc., Canada) of $O_2$ (300 sccm of 99.999 % $O_2$ in $N_2$, Daedeok Gas Co. Ltd.). $NO_2$ (2 % in $N_2$, Daedeok Gas Co. Ltd.) was added in two different positions (500 and 200 sccm, respectively) into a quartz reactor (5 cm i.d. and 50 cm length) for

efficient production of $N_2O_5$ crystals. To minimize $HNO_3$ formation in the reactor, any $H_2O$ which can be present in the gas supplies as well as on all surfaces in the generation system was removed by heating the reactor (up to 120 ℃) before the injection and by applying dry ice traps in front of the reactor during the synthesis. After the synthesis, additional $O_3$ was introduced on white $N_2O_5$ crystals to flush out the remaining $HNO_3$ for at least 30 minutes. The crystals were used immediately or trapped with dry ice and stored at -78 ℃ for later use.

Sub-ppbv to a few tens ppbv of $NO_3$ was produced by thermal equilibrium with the sublimated $N_2O_5$ introduced with a small flow of dry ZA (15 - 40 sccm) as a carrier gas passing the trap. Unlike previous studies (Fuchs et al., 2008; Kennedy et al., 2011; Odame-Ankrah and Osthoff, 2011; Wang et al., 2015; Wu et al., 2020), we did not provide any additional heat to shift the equilibrium towards $NO_3$ because the amount of $NO_3$ through this method was large enough to cover the ranges of typical atmospheric $NO_3$ mixing ratios in urban night conditions.





### 3.3.2 NO₃ transmission efficiency, $T_{NO_3}$

NO₃ losses ($T_{NO_3}$) of individual parts along the airway before the detection region (i.e. overflow inlet, filter, and cavity cell) were quantified via continuous injection (at least 5 minutes) of the synthesized NO₃ under dark condition. Differences in NO₃ concentrations with and without each part of the flow system were acquired by periodic switching between two conditions under the same residence time using a three-way solenoid valve. In order to determine the effect of NO₃ loss on the aerosol-accumulated filter, filters with total ambient suspended particle loadings of 203, 335, and 1010 μg·cm⁻³ were compared with a clean one.

Figure 6 presents the results of the $T_{NO_3}$ experiments for each of the test components with relative NO₃ concentration changes (concentrations in each step are normalized by the maximum value for every experiment resulting ranges from 745 pptv to 169 ppbv) since slow but steady increase in NO₃ concentrations was observed for all the experiments. We presumed that this drift may have been mainly due to the changes in temperature in dry ice bath where N₂O₅ crystals were placed and/or the variations in contact of ZA with the crystal surface. The comparisons of the retrieved NO₃ with and without each test part were achieved after reflecting the changes in slow increase by linear interpolations for the concentrations acquired in the same condition (shown as light red and gray dashed lines in Figure 6).

The $T_{NO_3}$ of individual parts are 98.9 (±1.9, 1σ), 88.1 (±2.6) and 93.1 (±0.3) % for coaxial overflow inlet (residence time < 0.03 seconds), cavity cell (residence time < 2.59 seconds), and filter assembly (clean case), respectively. The largest loss was observed within the cavity cell due to its relatively large surface area and long residence time compared to the other parts of the flow system. The quantified NO₃ loss on a clean filter surface (93.1±0.3 %) was similar to previous studies (85 (±10, 1σ) % on Aldener et al. (2006); 93 (±2) % on Dubé et al. (2006); 95 (±2) % on Fuchs et al. (2008); 84.8 (±10) % on Schuster et al. (2009); 92 (±3) % on Wang et al. (2015)). Interestingly, the used filters showed no significant differences compared to the clean one regardless of the ambient aerosol loadings within the experimental range (93.1 (±0.1, 1σ), 92.9 (±0.1), and 92.7 (±2.0) % for 203, 335 and 1010 μg·cm⁻³, respectively) which agree with Fuchs et al. (2008) and Zhou et al. (2018) . From the results, total $T_{NO_3}$ was quantified to be 81.2 % (±2.9, 1σ) under 2.5 slpm sampling condition.

### 3.4 Linearity tests

To test the linearity in signal response against the concentrations of species, standard injection experiments were performed. Multiple mixing ratios were achieved by regulating the degree of dilutions in synthesized N₂O₅ crystals, NO₂ standard (5 ppmv in N₂, KRISS, uncertainty: 1 %), and deionized water from the bubbler. As described in section 3.3.2, a slow and steady increase in NO₃ was observed, varying from 746 to 1045 pptv under the constant dilution ratio of 1:150 from beginning to end of the experiment. For tracking this drift in NO₃ standard, we alternated various dilution conditions with the base one (dilution ratio set as 1:150, shown in black markers in Figure 7 (a, b)) and applied the linear interpolation of retrieved NO₃ concentrations in those conditions.



Figure 7 shows the results of standard additions with respect to the elapsed time (left) and other independent concentration evaluation parameters (right; i.e. dilution ratio for $NO_3$, nominal concentration for $NO_2$, and independent RH measurement for $H_2O$). Dashed lines in Figure 7 (b, d, and f) represent the correlations considering their errors (uncertainty of the parameter on x-axis and $1\sigma$ variabilities in measurements on y-axis). Dilution ratio of $NO_3$ is calculated from the flow rate of ZA passed

over the source divided by the total flow rate and $NO_2$ standard concentration is from nominal concentration on manufacturer's specification with dilution ratio. RH was measured at the inlet tip by the humidity transmitter (HMT337, Vaisala, Finland) with the measurement uncertainty of 1 %.

All species show good linearities ($R^2$ of 0.9918, 0.9985, and 0.9980 for $NO_3$, $NO_2$, and $H_2O$, respectively) indicating the feasibility of atmospheric applications on those species. For $NO_2$, the intercept of 0.12 ppbv is insignificant considering the

limit of detection (will be discussed in Sect. 3.5). However, the intercept of -143 pptv in $NO_3$ is larger than the observed precision of the instrument (Sect. 3.5). It is likely due to the variabilities in our $NO_3$ source and/or the variations in offsets of slow drift correction since the source has the minimum flow rate requirements to operate. In addition, the negligible retrieved $NO_3$ (0.077±1.46 pptv, average and $1\sigma$ for 1s integration data) during the $N_2$ injection experiment, described in Sect. 3.5, can be used as an alternative to evaluate the zero offsets.

For $H_2O$, 84 ppmv of the intercept was found which is in a similar order of magnitude as with its precision (35 ppmv for 1 s integration time). This may be attributable to not only the random noise in detection but also the zero offset in the humidity transmitter (uncertainty: 1%). From this high linear response of our instrument in $H_2O$ measurements in varying atmospheric relevant ranges, we would like to emphasize that the difficulties in retrieval in $NO_3$ measurement due to $H_2O$ can be alleviated by simultaneous measurement with instrument-specific absorption spectrum of $H_2O$ without any pre-treatment to remove it.

**3.5 Precision and accuracy**

Allan deviation method is often used to determine the instrumental precision and the optimal integration time (Allan, 1966; Werle et al., 1993). Minimum detectable extinction for each pixel of the CCD was extracted from 1 hour injection of UHP $N_2$ with 1 second integration time. Figure 8 (a) shows the time series of the light extinction ($\alpha_{N_2}$) at 662 nm (corresponding to 1024th pixel) where $NO_3$ absorbance peaks. $\alpha_{N_2}$ shows no significant time-dependent changes and it deviates around zero,

which is likely dominated by white noise. Figure 8 (b) shows the Allan deviation for the single pixel corresponding to 662 nm. Up to around 900 s, $1\sigma$ precision generally follows the statistical limit which implies that there are no significant integration time-dependent systematic errors up to 900 s but it starts to gradually diverge after that, which is likely due to the instrumental drifts such as changes in conditions of the light source and/or CCD.

By only using the corresponding absorption cross-section on that single pixel, the detection limits for $NO_3$, $NO_2$, and $H_2O$

are determined to be 1.41 (0.15) pptv, 6.92 (0.73) ppbv, and 35.0 (3.69) ppmv, respectively, for 1 (60) second(s) averaging under 1 atm and 25 ℃ condition, by following Fouqueau et al. (2020). However, one should note that the spectral retrievals





through the optimized fitting algorithm are likely to produce even lower detection limits than those from the single pixel because this method relies on the absorption features on broad wavelength ranges among hundreds of pixels.

Measurement uncertainties (1σ) for NO₃, NO₂, and H₂O are calculated to be 10.8, 5.2, and ≥20.5 %, respectively, by
Gaussian propagation of the errors in absorption cross-section (NO₃: 10 %, NO₂: 3.3 %, and H₂O: ≥20.1 %), effective cavity length (3.4 %), and HR mirror reflectivity (2.2 %). Note that the fitting errors are not included here because the mathematical error varies with target species abundance. If there are strong signatures of target species in the measurement, fitting errors are negligible. However, for the extremely low abundance condition, the absorption features of target species weaken and thus the fit result likely end up to produce physically meaningless number with large error because of the limitations in numerical
fitting algorithms. Hence, when the fitting error outweighs the abundance of target species, interpretations should be limited.

Table 1 summarized the cavity characteristics and performances of the existing BBCEASs for NO₃ measurement. Our instrument has the longest effective light path length even though the mirror displacement is relatively short. Since this system is able to observe the optical extinctions in the order of $10^{-10}$ cm$^{-1}$ within 1 second time resolution, we can conclude that our instrument is adequate for measuring ambient NO₃ abundances in terms of sensitivity aspect. The capabilities of operation and
utilization for actual application are described in the following section.

## 4 Field deployment

In order to demonstrate the feasibility of the instrument in field measurements, we deployed our system on the Korean ice breaker R/V *Araon* and operated from late July to early September in 2021 for the expedition in Chukchi Sea and East Siberian Sea of Arctic Ocean (Figure 9 (a)). The instrument was housed in a seatainer placed on the compass deck (29 m above sea
level). Inlet was installed on the window and covered by the weatherproof-designed stainless steel pipe (7.5 cm o.d.). To minimize the loss of NO₃ along the sampling line, air was subsampled from the center of main flow (1 in. o.d., 1 mm thickness, PFA tubing, 20 slpm). The profile of the main flow was maintained to be steady and laminar (Reynolds number ≅ 1230) by the blower (DB-200, Manseung Electric Co., Korea). Total length and residence time inside the main flow was kept to be as short as possible to minimize the loss. However, due to the physical limitation of the instrument placement in the seatainer,
the length of subsampled PFA tubes was elongated (length: < 1 m, residence time: < 1.5 s) and the total transmission efficiency of NO₃ for this deployment was decreased by 65.1 % (±2.14 %, 1σ).

Aerosol filters were replaced by an integrated auto-filter changer (Dubé et al., 2006) only during the early and later stage of the mission near the coastal region of Northeast Asia for every three hours and manually changed with intervals of four to six hours in remote region since the demands for changing the filter were scarce due to the low aerosol loading in the Arctic
region. The $R(\lambda)$ and $I_{out,0}(\lambda)$ were checked every two hours and instantaneously interpolated for real-time $\alpha(\lambda)$ calculation. During the campaign, $R$(at 662 nm) were varying in the range from 99.9985 % to 99.9989 %, which are lower than the best performances of the instrument in the laboratory, mainly due to the difference in proficiency and environment for cleaning



optics but still high enough for ambient monitoring of $NO_3$. The negligible change in $R$ is a direct evidence that our vibration-resistant design is robust despite the strong vibrations in the platform due to the sea ice breaking activities.

Figure 9 (b-d) shows the time series of $H_2O$, $NO_2$, and $NO_3$ (1 minute averaged), as well as $O_3$ monitored by the UV absorption instrument (49i, Thermo Fisher Scientific, USA), radiance (CNR4, Kipp & Zonen, Netherland) and production rate of $NO_3$, $P(NO_3)$, calculated as equation (3). For $H_2O$, our measurements were compared to the calculated values from the pressure (PTB110, Vaisala, Finland), relative humidity, and temperature (HMP155, Vaisala, Finland) data measured on mainmast of the icebreaker. Here we only show the selected period (23 August at 17:00 – 25 August at 15:10 in 2021 UTC)

when the $NO_3$ signals were continuously observed well above the detection limit. $H_2O$ mixing ratios measured by our instrument were ranged from 4160 to 6510 ppmv (mean of 5580 ppmv) and average (maximum) value(s) of $NO_2$ and $NO_3$ were 3.21 (23.9) ppbv and 2.53 (9.51) pptv, respectively. $H_2O$ concentrations measured by both instruments were in good agreement considering the uncertainty of $H_2O$ for our instrument ($\geq$ 20.5 %).

$$P(NO_3) = k_{R1} [NO_2] [O_3] \tag{3}$$

During the campaign, several fresh emissions from R/V *Araon* were observed, represented by sharp changes in $O_3$ and $NO_2$ as shown in Figure 9 (c) indicating that $O_3$ was titrated by NO and formed $NO_2$. These exhaust emissions lasted from a few minutes to several hours depending on the atmospheric conditions such as wind direction and atmospheric stability as well as movement of the vessel (i.e. sailing or anchoring in one place for marine survey).

P(NO_3$) was generally small but rose up to 0.66 ppbv hr$^{-1}$ (mean of 0.21 ppbv hr$^{-1}$) depending on whether the sampled air

masses were directly influenced by the ship plumes or not, but still lower than those observed in previous works during summertime; 0.31 ppbv hr$^{-1}$ in rural areas (Geyer et al., 2001a) and 1.10 - 3.2 ppbv hr$^{-1}$ in urban areas (Wang et al., 2013; Brown et al., 2017; Zhou et al., 2018) . Within the plume condition, maximum $NO_3$ increased up to 1.85 pptv even though there was sunlight at that time. The trend of $NO_3$ concentration was well-matched with $P(NO_3)$ for most of the period indicating the suitability of our instrument for ambient $NO_3$ measurements. Further analysis related with regional impacts of ship plume

chemistry together with $NO_3$ oxidation assessment with observed VOCs and DMS would be interesting topics for future studies.

## 5 Conclusions

This paper describes our newly built cavity-enhanced absorption spectrometer for simultaneous measurements of ambient $NO_3$, $NO_2$, and $H_2O$ from their absorption features in 659.28-671.94 nm. High performances in measurement capabilities and simplicity in maintaining and processing schemes were achieved by applying high-reflection mirrors (up to 99.9995 % at 662

nm), by integrating the mirror purge and cage system as well as by simultaneous quantification of $H_2O$ using its measured spectrum. Generally, the light at 662 nm travels more than 40 km (up to 101.5 km) within the compact cavity cell (51.5 cm), which enables sensitive measurements of the target species. To overcome the difficulties in $H_2O$ treatment for accurate $NO_3$ measurement, the measured absorption spectrum of $H_2O$ was used and our instrument showed high linearity for varying



atmospheric relevant ranges of $H_2O$. The transmission efficiency of $NO_3$ from the inlet tip to the detection region was evaluated
as 81.2 ($\pm$2.9, 1$\sigma$) % within the residence time of 2.59 seconds from the prepared $NO_3$ addition experiments. Consequently,
for $NO_3$, $NO_2$, and $H_2O$, the measurement accuracies and 1$\sigma$ detection limit for 1 second integration time for a single pixel
CCD were determined as 10.8 %, 4.7 %, and $\geq$20.5 % with 1.41 pptv, 6.92 ppbv, and 35.0 ppmv, respectively, which are
sufficiently low for ambient applications.

The instrument was successfully deployed aboard the Korean ice breaker R/V *Araon* and captured not only the background
condition of the atmosphere over the open ocean in the Arctic but also the highly structured features of the plumes which
originated from the vessel exhaust during the campaign. In addition, the trend of $NO_3$ concentration was well-matched with
the calculated $P(NO_3)$ which serves as a proof of the potential for active applications of this instrument in further studies not
only in urban regions but also in pristine regions with any mobile platforms including aircraft and research vessel.

**Data availability**

The datasets used in this study are available upon request to the corresponding author, Kyung-Eun Min (kemin@gist.ac.kr).

**Author contributions**

WN and KEM contributed to designing and performing this study. TSR provided the data for $O_3$ and helped for the field
deployment. WN and KEM wrote the manuscript with contributions from CC and BP. All co-authors revised the content of
the original manuscript and approved the final version of the paper.

**Competing interests**

The authors declare that they have no conflict of interest.

*Acknowledgements.*

We thank all the crew members of R/V *Araon* for their support on the mission.





**Financial support**

This work was supported by the Korea Polar Research Institute (KOPRI) (PE21900) and by Korea Environment Industry & Technology Institute (KEITI) through Public Technology Program based on Environmental Policy Project, funded by Korea Ministry of Environment (MOE) (2019000160004).

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



**Table 1:** Comparison of BBCEAS performances for NO₃ measurement.

| Reference | Eff. light path length (mirror displacement) | Reflectivity (max. performance) | Detection limit (time resolution) | Accuracy | Application[a] |
|---|---|---|---|---|---|
| Ball et al. (2004) | N/A | 99.9965 % @ 670 nm | 2.5 pptv (1σ, 516 seconds) | N/A | Laboratory |
| Venables et al. (2006) | 2 km (4.5 m) @ 665 nm | 99.775 % @ 665 nm | 4 pptv (N/A,60 seconds) | 14 % | Laboratory |
| Langridge et al. (2008) | 11.8 km (1.1 m) @ 660 nm | 99.9913 % @ 660 nm | 0.25 pptv (1σ ,10 seconds) | N/A | M, France |
| Varma et al. (2009)[b,c,d] | 33.5 km (8.6 m) @ 665 nm | 99.98 % @ 662 nm | 1 pptv (1σ ,5 seconds) | 16 % | M, Ireland |
| Kennedy et al. (2011) | 10 km (0.94 m) | N/A | 1.1 pptv (1σ, 1 second) | 11 % | M, UK |
| Wu et al. (2014) | 22 km (2 m) | 99.991 % @ [638, 672 nm] | 7.9 pptv (N/A, 60 seconds) | 12 % | Laboratory |
| Wang et al. (2017)[d] | 6.13 km (0.33 m) @ 662 nm | 99.9936 % @ 662 nm | 2.4 pptv (1σ, 1 second) | 19 % | U, China |
| Suhail et al. (2019)[b] | 6.5 km (4.5 m) @ 660 nm | 99.95 % @ 660 nm | 36 pptv (N/A, 600 seconds) | N/A | SU, China |
| Wang and Lu (2019)[b] | 5.1 km (0.84 m) @ 665 nm | 99.985 % @ 662 nm | 3.0 pptv (2σ, 30 seconds) | 11–15 % | U, China |
| Fouqueau et al. (2020) | 3.15 km (0.82 m) @ 662 nm | 99.974 % @ 662 nm | 6 pptv (N/A, 10 seconds) | 9 % | Laboratory |
| This work | 101.5 km (0.52 m) @ 662 nm | 99.9995 % @ 662 nm | 1.41 pptv (1σ, 1 second) | 10.8 % | M, Arctic |

a: U : urban region, SU : suburban region, M : marine region

b: systems with open cavity

c : ambient NO₃ was well below the detection limit through the whole measurement period

d: average values of reflectivity are noted instead of maximum values




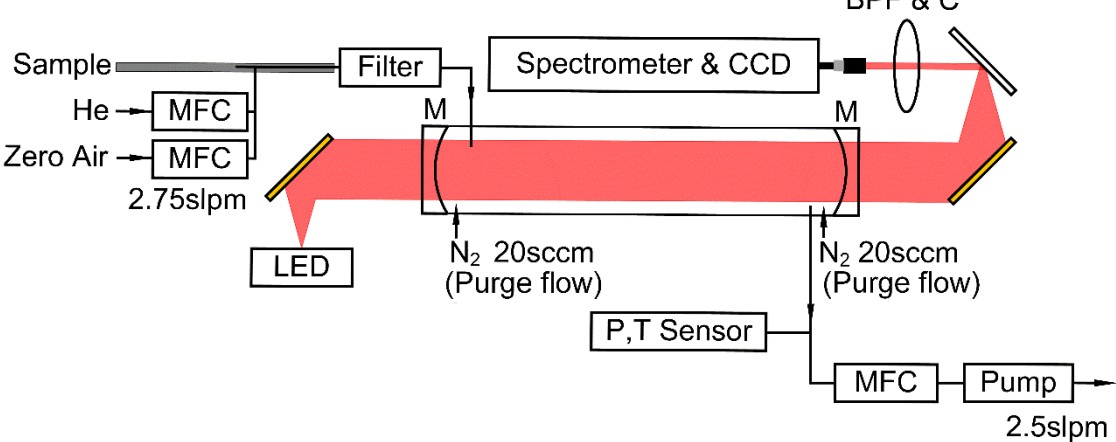

**Figure 1:** Schematic of the NO₃ BBCEAS system with light path (shaded in red) and gas flows (black arrows). High-reflection mirrors (M), band-pass filter (BPF), fiber collimator (C), mass flow controllers (MFC), and sensors of precision pressure transducer (P) and thermocouple (T) are marked.

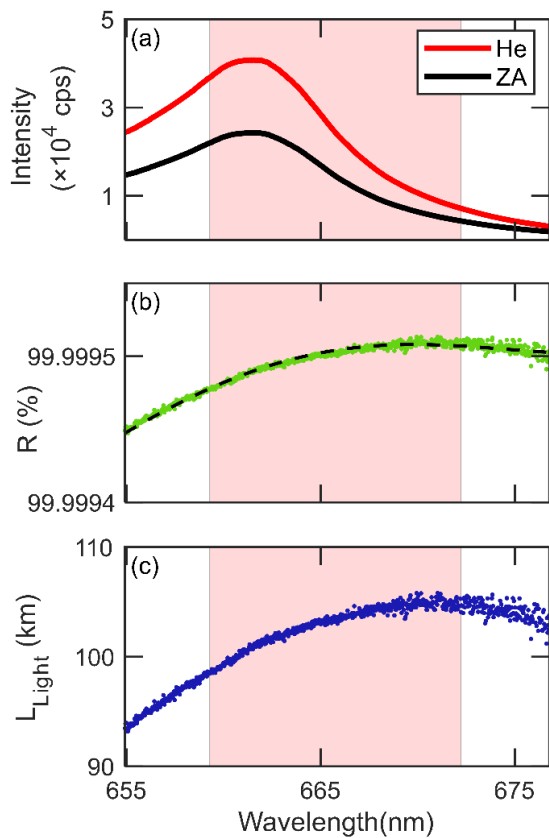

**Figure 2:** Cavity characteristics of 1 minute averaged **(a)** transmitted light intensities of zero air (ZA) and He, **(b)** mirrors reflectivity, $R$ (dashed black line is fitted with fourth-order polynomial) and **(c)** effective light path length, $L_{Light}$. Shaded area in light red represents fitted range for the number density retrieval.




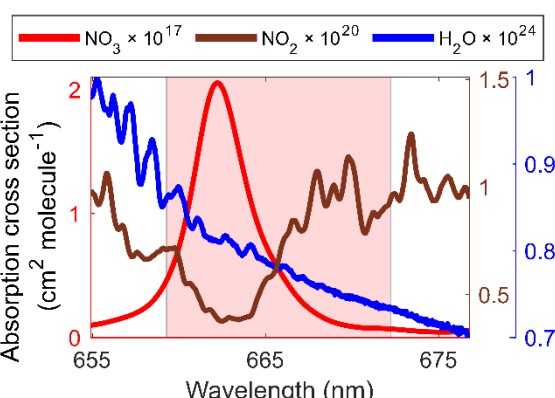

**Figure 3:** Used absorption cross-sections of $NO_3$, $NO_2$, and $H_2O$ and fitting range (shaded in light red) for number density retrievals.

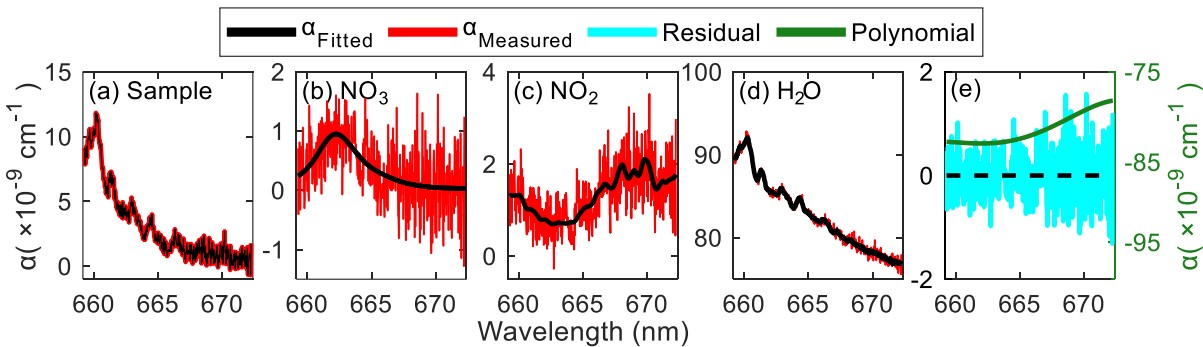

**Figure 4:** Spectral fitting example (2 seconds average) of ambient measured (red) and fitted (black) **(a)** total extinction, **(b)** $NO_3$, **(c)** $NO_2$, **(d)** $H_2O$ and **(e)** polynomial (green) with residual (cyan), respectively. Data was acquired on August 26th, 2021 (UTC) from shipborne measurement on open ocean in Arctic region.

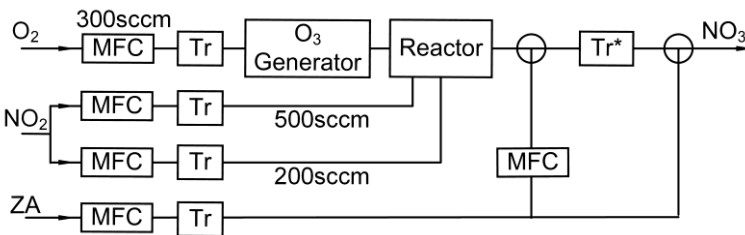

**Figure 5:** Schematic diagram of $NO_3$ generation system with flow paths (arrows). Dry ice cold trap for $H_2O$ removal (Tr) and for sample collection (Tr*) are also shown.





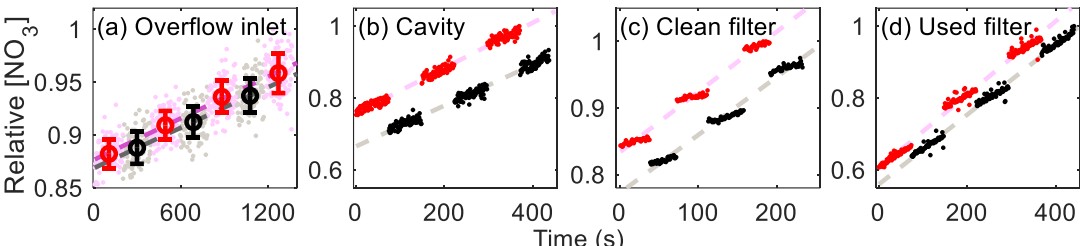

**Figure 6:** Relative NO₃ concentrations with (black) or without (red) test sections in **(a)** overflow inlet (averages as circle and 1σ as error bars), **(b)** cavity, **(c)** clean and **(d)** used filter (aerosol loading of 1,010 μg·cm⁻³). Light red and gray dashed lines represent concentration drifts inferred from linear interpolation of each condition.

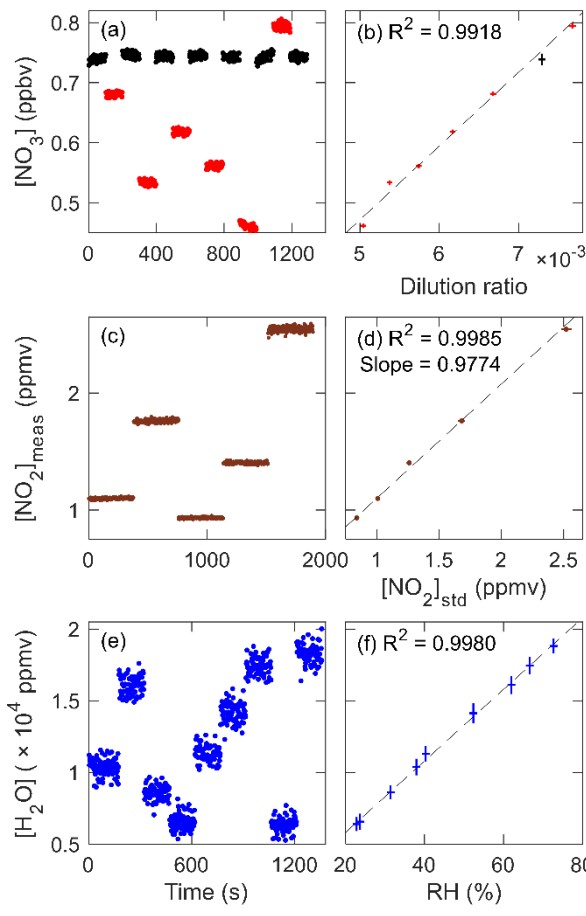


**Figure 7:** NO₃ **(a, b)**, NO₂ **(c, d)**, and H₂O **(e, f)** mixing ratios with elapsed times and other independent abundance evaluators in standard addition experiments. For NO₃, correction of the steady drift in NO₃ source bath was applied by linear interpolation of data with frequent injections of constant dilution condition (black). Error bars in (b, d, and f) represent 1σ variabilities for 2 seconds integration data (vertical) and uncertainty of evaluators (horizontal), while dashed lines show linear correlations.




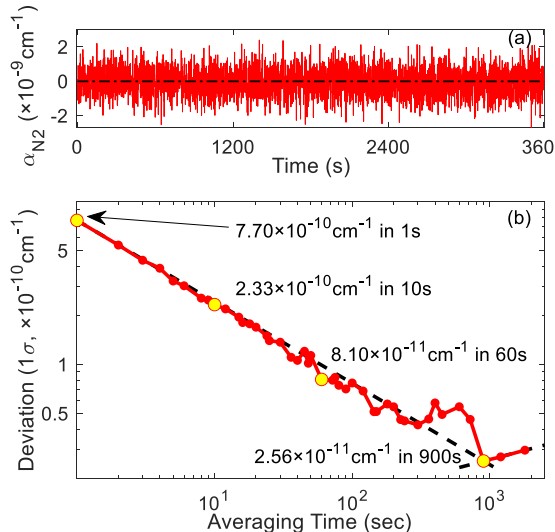

**Figure 8: (a)** Time series of light extinction for 1 hour of $N_2$ injection (1 second integration) and its **(b)** Allan deviations ($1\sigma$) for the single pixel at 662 nm. Yellow circles represent deviations for 1, 10, 60 and 900 seconds, respectively, and dashed line indicates theoretical limit representing white noise (slope : -0.5)


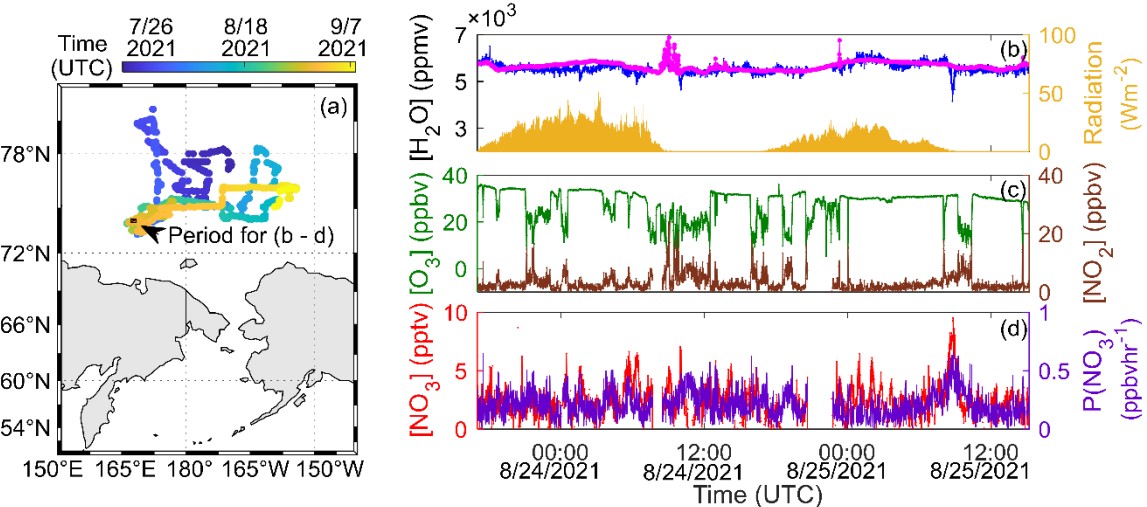

**Figure 9: (a)** A map with cruise track of R/V *Araon* and **(b–d)** representative time series of 1-minute averaged $H_2O$ (measured by our instrument on compass deck in blue and a Vaisala instrument on mainmast in pink), radiation, $NO_2$, $O_3$ and $NO_3$ with calculated production rate of $NO_3$ for 23 August at 17:00 – 25 August at 15:10 in 2021 (UTC) on the open ocean in Arctic region.
