# Peer review of "Development of a broadband cavity-enhanced absorption spectrometer for simultaneous measurements of ambient $NO_3$ , $NO_2$ , and $H_2O$"

_EGUsphere, 2022_

## Author Response (AR1)

**Response to Reviewers**

**Manuscript Number: egusphere-2022-145**

**Manuscript Title : Development of a broadband cavity-enhanced absorption spectrometer for simultaneous measurements of ambient $NO_3$, $NO_2$, and $H_2O$**

**The discussion below includes the complete text from the reviewer, our responses to each comment, and the corresponding changes made in the revised manuscript.**

**All the line numbers refer to the original manuscript and our responses (blue) along with changes in the revised manuscript (red) to the comments are color coded for your convenience.**

**Response to Reviewer #1 Comments:**

This paper reported a newly developed IBBCEAS system to measurement $NO_3$, $NO_2$ and $H_2O$ near 662 nm, in which the detection of $NO_3$ with high accuracy is the key target. The non-linear absorption of water vapor near 662 nm is a large interference and lead to the retrieval of weak $NO_3$ absorption challenging. Several studies have been trying to address this issue. In this study, the design of the optical system was adopted from a well-established instrument for measuring the glyoxal and nitrous acid (Min et al., 2016), and added a purging flow on high-reflection mirror surfaces. I believe the novelty of work is making effort to retrieve the ambient absorption of water vapor by establish $H_2O$ absorption cross section by the instrument measurement in advance. Overall, this topic is within in the scope of AMT, and this manuscript is well written with a comprehensive characterization in the lab as well as a good performance in the field test. The authors did a good job, I would like to recommend this paper to be published subject to a minor revision.

We would like to thank the reviewer for the positive review and helpful comments.

**General comments:**

Line 187-195, I am very confused how did you established the $H_2O$ absorption cross section. Are you measured the water absorption at a certain RH or a series of RH level at room temperature? We know that the temperature and pressure in the detecting tube would influence the water absorption cross section, how to deal with these variations in ambient conditions? Given the importance of this issue, I suggest the authors provide more details about it in the revised version.

We measured the $H_2O$ absorption spectrum under the constant condition of *22.7±0.1 °C, 991.5±0.1 hPa, and* 12.34±0.05 % RH condition (average±1σ).

*Based on the comparison test among fit residuals with different RH conditions during the linearity test described in Sect. 3.4,* no statistically significant difference (one-way analysis of variance, ANOVA, p-value = 0.4698) was observed. Together with the high linearity between retrieved $H_2O$ abundance and RH, thus, we can conclude that the possible changes of absorption cross-section of $H_2O$ under varying atmospheric relevant RH is undetectable with our instrument capability.

We agree on the concern of changes in $H_2O$ absorption cross-section with varying temperatures. Thus, we took an active temperature control strategy, especially for all the optic components and the cavity, by integrating an optic box to maintain constant temperature during a field mission. The temperature of the optic box was controlled to be constant by thermoelectric assemblies (Laird Thermal Systems, Inc.) composed of TECs, heat sink blocks, and fans. From the advantage of active operation of the optic box, the temperature of the sample in the cavity was 22.02±0.90 °C (average±1σ) throughout the field campaign. The difference in $H_2O$ absorption cross-section spectrum due to changes in temperatures of the cavity for ambient sample measurement and *$H_2O$ absorption spectrum measurement was indistinguishable with the precision and uncertainty of our instrument.*

Also, the range of pressure in the cavity during the Arctic mission was 997.6±25.9 hPa (average±1σ). The effect of pressure variation in $H_2O$ cross-section is not measurable with our instrumental capability. However, this effect can be crucial when the pressure of the sample is significantly different with the cross-section experimental condition such as airborne observation and it needs to be considered as the reviewer mentioned.

To clearly address these points, we have added information to the text:

Line 110: "… lines (NE-2, Ocean Optics Inc., USA). The entire optical layout was housed in a temperature-controlled optic box to maintain constant performances regardless of environmental temperature changes."

Line 190-192: "For that, $H_2O$ was  injected via the constant flow of ZA through a deionized water bubbler at  temperature of 22.7 °C and pressure of 991.5 hPa with 12.3 % relative humidity as averages, while activated carbon denuder (6 mesh, Ecotech Pty Ltd., Australia) as well as Drierite filter (8 mesh, Thermo Fisher Scientific, USA) were installed on upstream of the bubbler to remove possible contaminants in ZA cylinder."

Line 329: "… due to the sea ice breaking activities. During the mission, the averages (±1σ) of temperature and pressure of the sample in the cavity were 22.02 (±0.90) °C and 997.6 (±25.9) hPa. And the changes in absorption cross-sections due to these variations were too small to be detected by our instrument."

How about the influence of the mirror reflectivity change in the water cross section? If the R decreased to 0.99999 for example, the previous measured water cross section still working?

We think that the cross-section is mainly affected by wavelength resolving power (rather than $R$), which is determined by the characteristics of the detection parts (i.e. entrance slit width, groove number, and angle on the grating surface of the spectrometer, and full well depth of CCD). For our instrument, the detection parts are designed to hold constant conditions, and therefore changes in cross-section are hardly expected unless $R$ becomes too low to detect the ambient signals. With an aid of $N_2$ purging system, we have not experienced remarkable $R$ degradation. The range of $R$ during the Arctic mission was 99.9985 - 99.9989 %, as mentioned in line 326.

How about the temperature range in the field campaign in Arctic regions, is it possible lead to a bias in retrieving $H_2O$ absorption?

As replied in 1st general comments, with an aid of optic box, the cavity temperature was 22.02±0.90 °C (average±1σ), even though the ambient temperature in the Arctic varies from -4.76 °C to 1.78 °C (-1.02 °C as average). In addition, the temperature of the seatainer was also well-controlled by an air conditioner.

We have added information to the text:

Line 314-315: "The instrument was housed in a temperature-controlled seatainer placed on the compass deck (29 m above sea level)."

Line 320, how the transmission efficiency determined in the field campaign, especially the loss in the sampling tube, is it scaled by the residence time in this part?

We measured the transmission efficiency after the campaign and have added this information to the text:

Line 320-321: "… and the total transmission efficiency of $NO_3$ for this deployment was decreased by 65.1 % (±2.14 %, 1σ, quantified by post-campaign experiment through the same method as described in Sect. 3.3.

."

**Technical corrections:**

Line 240, the unit of aerosol loading should be µg rather than µg cm$^{-3}$? Please clarify it.

We thank the reviewer for the correction. We have edited the text:

Line 239-241: "In order to determine the effect of $NO_3$ loss on the aerosol-accumulated filter, filters with total ambient suspended particle loadings of  2480, 4075, and 12308 µg  were compared with a clean one."

Line 254-256: "Interestingly, the used filters showed no significant differences compared to the clean one regardless of the ambient aerosol loadings within the experimental range (93.1 ($\pm$0.1, 1$\sigma$), 92.9 ($\pm$0.1), and 92.7 ($\pm$2.0) % for  2480, 4075, and 12308 µg , respectively) which …"

Line 789-791: "**Figure 6**: Relative $NO_3$ concentrations with (black) or without (red) test sections in **(a)** overflow inlet (averages as circle and 1$\sigma$ as error bars), **(b)** cavity, **(c)** clean and **(d)** used filter (aerosol loading of 12308 µg ). Light red and gray dashed lines represent concentration drifts inferred from linear interpolation of each condition."

Line 146-147, typo error.

Revised.

The dot size in figure 7(b, d, f) is too small.

We have changed the marker of the data from the dot with the cross to the oval and therefore the data average and their errors are easily observed together in Fig. 7 (b, d, f).

[Figure]

Line 793-796: "**Figure 7:** NO$_3$ **(a, b)**, NO$_2$ **(c, d)**, and H$_2$O **(e, f)** mixing ratios with elapsed times and other independent abundance evaluators in standard addition experiments. For NO$_3$, correction of the steady drift in NO$_3$ source bath was applied by linear interpolation of data with frequent injections of constant dilution condition (black).  In (b, d, and f) axes of ellipse represent 1σ variability for 2 seconds integration data (vertical) and uncertainty of evaluators (horizontal), while dashed lines show linear correlations."

**Response to Reviewer #2 Comments:**

**General Comments:**

This paper details the development of a new BBCEAS system for simultaneous measurements of the trace gases $NO_3$, $NO_2$, and $H_2O$. Unlike previously developed absorption-based sensors for $NO_3$, this study emphasizes the utility of retrieving the water vapor signal, which has strong absorption features in the detected spectral region around 662 nm, instead of correcting for water vapor as an interference in the $NO_3$ signal. The instrument demonstrates superior precision and comparable accuracy as compared to existing BBCEAS $NO_3$ measurements. Overall, this paper presents a thorough characterization and evaluation of the instrument performance and its field operation. It is well within the scope of AMT, and I recommend publication subject to the minor revisions detailed below.

We would like to thank the reviewer for the positive review and the useful comments. Our point-by-point responses follow with updated manuscript and supplementary materials.

**Specific Comments:**

1.  I agree with RC1 that the description of measuring the $H_2O$ absorption spectrum in the original text is unclear. I believe the authors have sufficiently addressed this concern in their response, as well as any concerns relating to temperature control of the instrument.

    We agree with the reviewer about the insufficient descriptions for the $H_2O$ absorption spectrum measurement and the temperature control of the instrument in the original manuscript. We have extended the description in the manuscript as written below. Please refer to our response in AC1 (Reviewer #1 General comment for Line 187-195).

    Line 110: "… lines (NE-2, Ocean Optics Inc., USA). The entire optical layout was housed in a temperature-controlled optic box to maintain constant performances regardless of environmental changes."

    Line 190-192: "For that, $H_2O$ was  injected via constant flow of ZA through a deionized water bubbler at  temperature of 22.7 °C and pressure of 991.5 hPa with 12.3 % relative humidity as averages, while activated carbon denuder (6 mesh, Ecotech Pty Ltd., Australia) as well as Drierite filter (8 mesh, Thermo Fisher Scientific, USA) were installed on upstream of the bubbler to remove possible contaminants in ZA cylinder."

    Line 329: "… due to the sea ice breaking activities. During the mission, the averages (±1σ) of temperature and pressure of the sample in the cavity were 22.02 (±0.90) °C and 997.6 (±25.9) hPa. And the changes in the absorption cross-sections due to these variations were small to be detected by our instrument."

2.  L207 states a "Fourth-order polynomial was applied to account for the optical drift and/or unaccounted extinctions such as absorption by ambient ozone." Was there any basis for selecting this functional form? The retrieval demonstrates that the polynomial fit is a quiet a large component of the overall signal. Please elaborate or clarify why this is the case.

    To our best knowledge, there are insufficient discussions on how it should be selected. In our case, an empirical decision was made to attain better fitting performances from laboratory and field measurements rather than just following the previous broadband cavity-enhanced absorption spectroscopy (BBCEAS) works that used fourth-order polynomial (Thalman and Volkamer, 2010; Thalman et al., 2015; Min et al., 2016; Washenfelder et al., 2016; Jordan et al., 2019; Barbero et al., 2020). More specifically, we decided the order of polynomial based on changes in fit coefficient uncertainty, root mean square (RMS) and chi-square of residuals.

For example, Figure S1 shows the residual RMSs, residual chi-squares, and fit coefficient errors of each species (normalized ones for simple comparison) as a function of polynomial orders for the data corresponding to the spectrum in Figure 4 in the main manuscript. Overall, the fitting with fourth-order polynomial shows better performance than the others. This exercise was repeated for different measurement environments.

We have added the sentence in the manuscript and Figure S1 in the SI:

Line 207-208: "Fourth-order polynomial was applied to account for the optical drift and/or unaccounted extinctions such as absorption by ambient ozone. Fit order was selected based on the resulted fitting statistics (i.e. fit coefficient uncertainties, root mean square and chi-square of residuals, Figure S1), which needs to be verified for different measurement applications."

[Figure]

**Figure S1:** Statistics of spectral fitting for ambient data with the order of the polynomial. Data was acquired on August 26th, 2021 (UTC) from shipborne observation, which matches with data in Figure 4. Normalized fit coefficient uncertainty which ranged from 0 to 1 was used for convenience.

3.  How reproducible are the NO₃ transmission results to the field environment? It seems this has been clarified in the author's response to RC1, but I'm curious if this would have to be characterized in each new environment.

    Based on the experiences we have with this newly built system, the transmission efficiencies of $NO_3$ ($T_{NO3}$) for all parts from the coaxial overflow inlet to the detection region hardly changed with varying aerosol loading within the experimental range (2480–12308 µg, described in Sect. 3.3.2). However, $T_{NO3}$ is sensitive to the residence time along the airway (Liebmann et al., 2017). Thus, we presumed that the $T_{NO3}$ verified in Sect. 3.3.2 can be used unless changes in residence time exist due to the modifications in length and configuration of the coaxial inlet part as well as the flow rate of the sample.

    However, individual mission has its own limitation in instrumental deployment, thus the inlet characteristics such as length, shape, and material can be changed. For this reason, $T_{NO3}$ for the individual setup should be quantified for every campaign. We think that on-site verification of $T_{NO3}$ is ideal, and regular base checks throughout a mission is recommended.

    In the case of Arctic mission, however, we were not able to quantify $T_{NO3}$ during the mission due to the logistical issue (consumable supplies were not allowed due to COVID-19). Therefore, $T_{NO3}$ of the weatherproof-designed inlet with elongated PFA tubing before the cavity was evaluated after the campaign in the laboratory.

    To address this concern, the text in the main manuscript has been edited as:

    Line 319-321: "However, due to the physical limitation of the instrument placement in the seatainer, the length of subsampled PFA tubes was elongated (length: < 1 m, residence time: < 1.5 s) and the total transmission efficiency of NO₃ for this deployment was  65.1 % (±2.14 %, 1σ), quantified by post-campaign experiments through the same method as described in Sect. 3.3."

4.  The description of the NO₃ dilutions in the linearity test are somewhat unclear. Where is the drift in the NO₃ concentration evidenced in Figure 7? Or have the data in Fig 7a,b already been corrected for the linear drift? Please be explicit as to what the red and black dots indicate in these figures. It is not stated in the text or in the figure caption.

    Data in Figure 7 (a and b) have already been corrected for the drift in the NO₃ source. For more clarity, we added Figure S2 which shows the data without the correction. NO₃ concentration under the constant dilution condition (black) was interpolated (gray dotted line in Figure S2 (a)) and used as the baseline for the drift of NO₃ standard. This baseline was subtracted from the data under both constant (black) and different (red) dilution conditions.

    We have added the phrases in the manuscript and Figure S2 in the SI:

    Line 261-265: "As described in section 3.3.2, a slow and steady increase in NO₃ was observed, varying from 746 to 1045 pptv under the constant dilution ratio of 1:150  throughout the experiment (Figure S2). For tracking this drift in NO₃ standard, we alternated various dilution conditions with the base one (dilution ratio set as 1:150, shown in black markers in Figure 7 (a, b)) and applied the linear interpolation of retrieved NO₃ concentrations in those conditions (gray dotted line in Figure S2 (a)). This baseline which depicts the changes in NO₃ source drift was subtracted from the data for both constant (black) and different (red) dilution conditions."

[Figure]

**Figure S2:** Time series of the linearity test for $NO_3$ **(a)** without the correction for the steady drift in $NO_3$ source (gray dotted line) and **(b)** with the correction (same as Figure 7 (a)).

[Figure]

Line 793-796: "**Figure 7:** $NO_3$ **(a, b)**, $NO_2$ **(c, d)**, and $H_2O$ **(e, f)** mixing ratios with elapsed times and other independent abundance  evaluation parameters in standard addition experiments.  For $NO_3$, data with constant (black) and varying (red) dilution conditions were corrected for the steady drift in $NO_3$ source by linear interpolation (Figure S2). In (b, d, and f) axes of ellipse represent 1σ variability for 2 seconds integration data (vertical) and uncertainty of each variable (horizontal) while dashed lines show linear correlations."

5. L321: The wording is unclear. Was the total transmission efficiency reduced by 65% of the lab-based value? Or reduced to a total transmission efficiency of 65%?

The total transmission efficiency was reduced to 65.1 %. To be clear, we have edited the corresponding words as we have replied in the 3rd specific comment.

**Technical Corrections:**

1. It would be helpful to see all the detection limits in Table 1 for the same integration time if possible (for ease of comparison).

We have checked the references in Table 1 to match the integration time for the detection limit by collecting all available information. In general, no explicit information were reported with another integration time, but a few papers (Langridge et al., 2008; Kennedy et al., 2011; Wang et al., 2017) provide Allan deviation plots which can be used to approximate their detection limits for the same integration time. However, due to the errors in reading from printed graphs, we included the approximated detection limits only in Table S1 and SI.
In regard to Table 1, we have corrected for the mistakes we made in describing the performance of Venables et al. (2006) and Varma et al. (2009); the integration time is 57 rather than 60 seconds in Venables et al. (2006) and the detection limit in Varma et al. (2009) is 2 rather than 1 pptv.

We have added the words in the manuscript and Table S1 in the SI:
Line 306: "Table 1 summarized the cavity characteristics and performances of the existing BBCEAS for $NO_3$ measurement. Inferred detection limits with the same integration time are available in Table S1."

**Table S1:** Comparison of BBCEAS performances for $NO_3$ measurement

| Reference | Reflectivity (max. performance) | Detection limit (time resolution) | | | Accuracy |
|---|---|---|---|---|---|
| | | reported | inferred[*] | | |
| Ball et al. (2004) | 99.9965 % @ 670 nm | 2.5 pptv (1σ, 516 seconds) | - | | N/A |
| Venables et al. (2006) | 99.775 % @ 665 nm | 4 pptv (N/A, 57 seconds) | - | | 14 % |
| Langridge et al. (2008) | 99.9913 % @ 660 nm | 0.25 pptv (1σ, 10 seconds) | 0.35 pptv (1σ, 5 seconds) | 0.1 pptv (1σ, 60 seconds) | N/A |
| Varma et al. (2009) | 99.98 % @ 662 nm | 2 pptv (1σ, 5 seconds) | - | | 16 % |
| Kennedy et al. (2011) | N/A | 1.1 pptv (1σ, 1 second) | 0.35 pptv (1σ, 5 seconds) | 0.1 pptv (1σ, 60 seconds) | 11 % |
| Wu et al. (2014) | 99.991 % @ [638, 672 nm] | 7.9 pptv (N/A, 60 seconds) | | | 12 % |
| Wang et al. (2017) | 99.9936 % @ 662 nm | 2.4 pptv (1σ, 1 second) | 0.6 pptv (1σ, 5 seconds) | 0.3 pptv (1σ, 60 seconds) | 19 % |
| Suhail et al. (2019) | 99.95 % @ 660 nm | 36 pptv (N/A, 600 seconds) | - | | N/A |
| Wang and Lu (2019) | 99.985 % @ 662 nm | 3.0 pptv (2σ, 30 seconds) | - | | 11–15 % |
| Fouqueau et al. (2020) | 99.974 % @ 662 nm | 6 pptv (N/A, 10 seconds) | - | | 9 % |
| This work | 99.9995 % @ 662 nm | 1.41 pptv (1σ, 1 second) 0.60 pptv (1σ, 5 seconds) 0.15 pptv (1σ, 60 seconds) | - | | 10.8 % |

[*]: Estimated detection limits from Allan deviation plot readings

**Reference**

Barbero, A., Blouzon, C., Savarino, J., Caillon, N., Dommergue, A., and Grilli, R.: A compact incoherent broadband cavity-enhanced absorption spectrometer for trace detection of nitrogen oxides, iodine oxide and glyoxal at levels below parts per billion for field applications, *Atmos. Meas. Tech.*, 13, 4317-4331, doi:10.5194/amt-13-4317-2020, 2020.

Jordan, N., Ye, C. Z., Ghosh, S., Washenfelder, R. A., Brown, S. S., and Osthoff, H. D.: A broadband cavity-enhanced spectrometer for atmospheric trace gas measurements and Rayleigh scattering cross sections in the cyan region (470–540 nm), *Atmos. Meas. Tech.*, 12, 1277-1293, doi:10.5194/amt-12-1277-2019, 2019.

Kennedy, O. J., Ouyang, B., Langridge, J. M., Daniels, M. J. S., Bauguitte, S., Freshwater, R., McLeod, M. W., Ironmonger, C., Sendall, J., Norris, O., Nightingale, R., Ball, S. M., and Jones, R. L.: An aircraft based three channel broadband cavity enhanced absorption spectrometer for simultaneous measurements of $NO_3$, $N_2O_5$ and $NO_2$, *Atmos. Meas. Tech.*, 4, 1759-1776, doi:10.5194/amt-4-1759-2011, 2011.

Langridge, J. M., Ball, S. M., Shillings, A. J. L., and Jones, R. L.: A broadband absorption spectrometer using light emitting diodes for ultrasensitive, in situ trace gas detection, *Rev. Sci. Instrum.*, 79, 123110, doi:10.1063/1.3046282, 2008.

Liebmann, J. M., Schuster, G., Schuladen, J. B., Sobanski, N., Lelieveld, J., and Crowley, J. N.: Measurement of ambient NO3 reactivity: design, characterization and first deployment of a new instrument, Atmospheric Measurement Techniques, 10, 1241-1258, doi:10.5194/amt-10-1241-2017, 2017.

Min, K.-E., Washenfelder, R. A., Dubé, W. P., Langford, A. O., Edwards, P. M., Zarzana, K. J., Stutz, J., Lu, K., Rohrer, F., Zhang, Y., and Brown, S. S.: A broadband cavity enhanced absorption spectrometer for aircraft measurements of glyoxal, methylglyoxal, nitrous acid, nitrogen dioxide, and water vapor, *Atmos. Meas. Tech.*, 9, 423-440, doi:10.5194/amt-9-423-2016, 2016.

Thalman, R., and Volkamer, R.: Inherent calibration of a blue LED-CE-DOAS instrument to measure iodine oxide, glyoxal, methyl glyoxal, nitrogen dioxide, water vapour and aerosol extinction in open cavity mode, *Atmos. Meas. Tech.*, 3, 1797-1814, doi:10.5194/amt-3-1797-2010, 2010.

Thalman, R., Baeza-Romero, M. T., Ball, S. M., Borrás, E., Daniels, M. J. S., Goodall, I. C. A., Henry, S. B., Karl, T., Keutsch, F. N., Kim, S., Mak, J., Monks, P. S., Muñoz, A., Orlando, J., Peppe, S., Rickard, A. R., Ródenas, M., Sánchez, P., Seco, R., Su, L., Tyndall, G., Vázquez, M., Vera, T., Waxman, E., and Volkamer, R.: Instrument intercomparison of glyoxal, methyl glyoxal and $NO_2$ under simulated atmospheric conditions, *Atmos. Meas. Tech.*, 8, 1835-1862, doi:10.5194/amt-8-1835-2015, 2015.

Varma, R. M., Venables, D. S., Ruth, A. A., Heitmann, U., Schlosser, E., and Dixneuf, S.: Long optical cavities for open-path monitoring of atmospheric trace gases and aerosol extinction, *Appl. Opt.*, 48, B159-B171, doi:10.1364/ao.48.00b159, 2009.

Venables, D. S., Gherman, T., Orphal, J., Wenger, J. C., and Ruth, A. A.: High sensitivity in situ monitoring of $NO_3$ in an atmospheric simulation chamber using incoherent broadband cavity-enhanced absorption spectroscopy, *Environ. Sci. Technol.*, 40, 6758-6763, doi:10.1021/es061076j, 2006.

Wang, H., Chen, J., and Lu, K.: Development of a portable cavity-enhanced absorption spectrometer for the measurement of ambient $NO_3$ and $N_2O_5$: experimental setup, lab characterizations, and field applications in a poll, *Atmos. Meas. Tech.*, 10, 1465-1479, doi:10.5194/amt-10-1465-2017, 2017.

Washenfelder, R. A., Attwood, A. R., Flores, J. M., Zarzana, K. J., Rudich, Y., and Brown, S. S.: Broadband cavity-enhanced absorption spectroscopy in the ultraviolet spectral region for measurements of nitrogen dioxide and formaldehyde, *Atmos. Meas. Tech.*, 9, 41-52, doi:10.5194/amt-9-41-2016, 2016.